

# Anesthesia videos in geriatric and elderly patients on YouTube: content, quality, reliability, and usefulness assessment

Turan Evran and Seher İlhan

Department of Anesthesiology and Reanimation, Pamukkale University School of Medicine, Pamukkale University, Denizli, Pamukkale, Turkey

## ABSTRACT

**Purpose**. This study aimed to assess the quality, reliability, content, and usefulness of YouTube videos related to anesthesia in geriatric and elderly patients.

**Methods**. Using Google Trends, the most popular search terms in the past five years, "geriatric anesthesia" and "anesthesia in the elderly," were identified. A search on YouTube using these terms initially yielded 200 of the most viewed videos for each term. After applying exclusion criteria, 87 videos were included for detailed analysis. Major themes and topics related to anesthesia in geriatric and elderly patients were identified using a pre-determined qualitative thematic analysis method. The usefulness of the videos was assessed using the specially developed Geriatric and Elderly Anesthesia Usefulness Score (GAEUS). The overall quality and reliability of the videos were evaluated using the Global Quality Scale (GQS) and the Modified DISCERN Scale (M-DISCERN), respectively. The average of the quality, reliability, and usefulness scores calculated by the researchers was used for consistency analysis.

**Results**. In our study, 48.3% (42) of the videos on geriatric and elderly patients concerning anesthesia on YouTube were created by personal blogs. The quality of the videos was measured using the GQS, with a mean score of 3.34 and a median of 3 (range: 1–5), showing no significant difference according to the video source ($p = 0.166$). Reliability was assessed using the M-DISCERN scale, with a mean score of 3.37 and a median of 3.50 (range: 1–5), again showing no significant difference according to the video source ($p = 0.097$). Usefulness was measured using the GAEUS score, with a mean score of 15.30 and a median of 12.5 (range: 2–63), which showed a significant difference according to the video source ($p = 0.000$). The average duration of videos with low usefulness was 31.59 minutes (range: 5–44), while the average duration of moderately and highly useful videos was 59.37 minutes (range: 19.44–119.05). This duration difference was statistically significant ($u = 2.569$, $p = 0.010$).

**Conclusion**. In our study, we examined YouTube videos covering anesthesia topics for geriatric and elderly patients. The highest usefulness scores were obtained from personal blogs; however, all sources generally showed low usefulness. The quality of the videos was assessed using the GQS, and their reliability was evaluated with the M-DISCERN. On both scales, the videos showed moderate performance across all sources. These findings indicate a need for more comprehensive and informative content on YouTube, especially for the education of healthcare professionals and patients. To better address the needs of elderly patients, the richness of content and educational value of these videos should be enhanced.

Corresponding author
Turan Evran, tevran@pau.edu.tr

## INTRODUCTION

Although there is no clear definition of old age in the literature, the World Health Organization defines individuals aged 60 and older as elderly (*Guidet et al., 2024*). By 2030, it is expected that one out of every six people will be over the age of 60 due to the aging population. At the same time, the proportion of elderly people requiring surgical intervention is increasing globally, and this group accounts for about half of the surgical population in developed countries (*Bettelli, 2023*). Compared with younger individuals, anesthesia-related mortality and morbidity rates are higher in geriatric patients. Advances in modern health have made it possible to perform surgery on elderly people more frequently and have increased the importance of anesthesia applications for this age group (*Al Harbi et al., 2023*). Advancements in primary and perioperative care have made anesthesia safer for geriatric patients. However, due to advanced age, these patients still carry a significant risk of morbidity and mortality (*Staheli & Rondeau, 2023*). The decrease in physiological functions and increased morbidity in elderly patients disrupt the recovery process after general anesthesia and increase the risk of postoperative complications. The aging process leads to pharmacokinetic and pharmacodynamic changes, affecting the interaction with anesthetic drugs and creating significant differences between age groups in drug responses (*Alghamdi, Almuzayyen & Chowdhury, 2023*). For this reason, geriatric patients with surgical requirements require a comprehensive assessment and care from the preoperative period to the postoperative period (*Aceto et al., 2020*). International health organizations encourage recommendations for perioperative care for geriatric and elderly patients, but studies in this area are insufficient (*Samuel et al., 2015*).

Today, health professionals, patients, and their families increasingly use the internet to find solutions to health problems, gain knowledge, and share their experiences (*Kartufan & Bayram, 2023*). The online health information-seeking behavior of older adults remains insufficiently explored. As the global population ages, the number of health-conscious elderly individuals is increasing, leading them to rely on online information sources to make informed health decisions (*Zhao, Zhao & Song, 2022*). Established in 2005, YouTube reaches 95% of internet users and has become an important source of education and information, offering rich video content. Although access to information provided over the internet is easy, the ability of users to effectively evaluate the quality and accuracy of this information is limited. This has led to an increase in studies on the quality and reliability of the information provided, especially on platforms such as YouTube (*Duran & Kizilkan, 2021*). It is crucial to evaluate the accuracy of the information presented in YouTube videos before directly applying it to patient management. To this end, verification by healthcare professionals, cross-referencing with reliable sources, and consideration of the potential risks of misinformation are necessary (*Kunze, 2020*; *Mylavarapu et al., 2023*; *Polat & Cankurtaran, 2025*). There are many videos on YouTube about anesthesia in geriatric

and elderly patients. However, comprehensive studies evaluating the content of these videos are not yet available in the literature.

The aim of this study is to assess the quality, reliability, content, and usefulness of anesthesia videos for geriatric and elderly patients on YouTube from the perspective of health professionals and patients, using the Global Quality Scale (GQS), the Modified Discern Scale (M-DISCERN), the Geriatric and Elderly Anesthesia Content Scale (GEACS), and the Geriatric and Elderly Anesthesia Usefulness Score (GAEUS). Our hypothesis is that anesthesia videos on YouTube for geriatric and elderly patients are of average quality, reliability, content, and usefulness for both health professionals and patients; however, significant differences exist depending on the source of the video.

## METHOD

Our study was designed as an observational research project that evaluates anesthesia videos related to geriatric and elderly patients on YouTube. Our research was approved by the Pamukkale University Faculty of Medicine Non-Interventional Clinical Research Ethics Committee on December 15, 2023 (No: E-60116787-020-462171).

### Keyword selection and video selection criteria

Our research was conducted within a systematic framework that included the selection of videos and data collection techniques. Both researchers set a date of January 20, 2024, for the identification and recording of the videos. The cache and cookies were cleared using the incognito mode of the Google Chrome browser. Google Trends is a tool that analyzes the popularity of specific keywords over time. In our study, Google Trends was used to identify the most searched terms over the past five years. After determining the two most used search terms, these terms were utilized to identify the videos to be analyzed on the YouTube platform.To select the appropriate videos for our study, we first used the terms 'anesthesia in elderly patients,' 'anesthesia in the elderly,' 'geriatric anesthesia,' 'anesthesia in geriatric patients,' and 'anesthesia in the elderly population' through Google Trends, where the most common search trends were deciphered. The most searched terms in the last five years were identified as 'geriatric anesthesia' and 'anesthesia in the elderly.'

Using these keywords, searches were made on YouTube. It was planned to include the top 100 most viewed videos for each search term in the study. The URLs and titles of the collected videos were saved in Microsoft Excel to protect against possible changes over time. After the registration process was completed, the accuracy and completeness of the data were checked. The YouTube videos included in the study were selected based on the highest number of views, and similar previous studies were referenced when determining the sample (*Kunze, 2020*; *Mylavarapu et al., 2023*; *Polat & Cankurtaran, 2025*).

Videos excluded from our study were determined based on specific criteria. Videos containing advertisements were not included, as they primarily serve promotional purposes for products or services. Duplicate videos, which represent multiple uploads of the same content, were excluded. Since the study focused solely on videos in the English language, non-English videos were omitted. Additionally, completely silent videos lacking speech or narration were not considered for evaluation. Non-original content, including videos that

were directly copied from another source and reuploaded, was also excluded. Lastly, videos in which the content did not match the topic stated in the title were not included in the analysis. During the YouTube search process, videos meeting these exclusion criteria were identified and excluded from the study.

## Analysis of videos and detection of subject content

The titles of the videos, the uploader's name, subscriber counts, and the uploader's country were recorded as basic information. The videos were categorized into three main types based on their source: videos from private and public hospitals and other healthcare institutions were classified as "health institutions"; universities and associations were labeled as "educational institutions"; and healthcare professionals, patients, personal video producers, and independent news channels without a corporate affiliation were categorized as "personal blogs".

The target audiences of the videos were determined to be patients and healthcare professionals based on the information in the channel's "about" section and the video content. Additionally, details such as the year of upload, view counts, days since upload, daily average view rates, video duration, and the number of likes and comments were also recorded. Data such as likes, comments, views, and upload date were collected on January 20, 2024. Since these metrics are time-sensitive, the reported values reflect only the moment of data collection. Multi-part videos were evaluated as separate videos for each segment.

Using a similar methodology employed by *Robb et al. (2022)*, the main topics and themes related to anesthesia in geriatric and elderly patients on YouTube were identified using a predetermined qualitative thematic analysis method.

In parallel with the studies of *Lim & Lee (2020)*, the main topic headings selected were preoperative, intraoperative, and postoperative management (*Kartufan & Bayram, 2023*). The researchers reached a consensus on a total of thirty-five themes under these headings, considering their 13–14 years of experience and daily practice.

Under preoperative management, nine themes were identified: definition and epidemiology, pathophysiology, frailty, American Society of Anaesthesiologists (ASA) score and general health assessment, nutritional status, comorbidities, alcohol and substance use, polypharmacy, and depression.

In intraoperative management, seventeen themes were identified, including anesthesia management, general anesthesia, spinal anesthesia, epidural anesthesia, peripheral nerve blocks, intubation, bag-valve-mask ventilation, extubation, monitoring, premedication, induction, anesthetic agents, neuromuscular blockers and antagonists, other pharmacological agents, positioning, fluid management, and blood transfusion. For postoperative management, nine themes were identified: analgesia and postoperative pain, respiratory complications, nausea and vomiting, hypothermia, mobilization, physiotherapy, long-term follow-up and management of chronic conditions, postoperative delirium, and postoperative cognitive dysfunction.

The specific content and scoring system used by *Qu et al. (2023)* was adapted for the YouTube videos in our study as the GEACS. To create the GEACS scale, it was first recorded whether each video contained information related to the predetermined themes.

To establish the GEACS scale, researchers first identified whether predefined themes were covered in each video. Each theme in the reviewed videos was scored as follows: 0 points if no information was provided, 1 point if superficial information was provided, and 2 points if sufficient and comprehensive information was included. Subsequently, the scores for all themes covered in a video were summed to calculate a GEACS score. Finally, the average of all theme scores assigned to a video was calculated to generate a single GEACS score for that video.

## Evaluation of the quality, reliability, and usefulness of videos

The overall quality of the videos was evaluated using the GQS and reliability was assessed with the M-DISCERN.

GQS evaluates a video's information flow, completeness, and usefulness for patients using a Likert scale ranging from 1 to 5 (*Aurlene et al., 2024*). In this scale, a score of one indicates poor quality, where the video has weak flow, low visual quality, lacks essential information, and provides no educational benefit to patients. A score of two represents low quality, meaning that while some information is present, the content is limited and offers minimal benefit to patients. A score of three indicates moderate quality, where some key information is adequately discussed, but the overall content is not comprehensive. A score of four represents good quality, with a well-structured flow, good visual and content quality, the inclusion of most relevant information, and significant patient benefits. A score of five indicates excellent quality, where the video is optimally structured, provides complete and comprehensive information, and is highly beneficial for patient education (*Baig et al., 2024*). In our study, videos scoring 4 or 5 were classified as high quality, those scoring 3 as moderate quality, and those scoring 1 or 2 as low quality.

The M-DISCERN scoring system is a method used to assess the reliability of health-related videos. It examines aspects such as clarity, source validity, information balance, the presence of additional reference sources, and whether controversial topics are addressed (*Kartufan & Bayram, 2023*). The modified version of this tool used in the study consisted of five yes/no questions, with each "yes" response assigned 1 point. Thus, the maximum possible score was 5. The evaluated questions were as follows: (a) Is the video clear, concise, and understandable? (b) Does it cite valid sources? (c) Is the provided information balanced and unbiased? (d) Does it offer additional sources for patient reference? (e) Does the video address controversial or ambiguous topics? Each "yes" answer received 1 point, while "no" received 0 points. In our study, videos with an M-DISCERN score above three were classified as good, those scoring exactly three as moderate, and those below three as poor.

The usefulness score developed by *Li et al. (2019)* was adapted for our study to create the GAEUS. To measure the educational usefulness of the videos, the GEACS scores were combined with the GQS scores to generate the GAEUS score. Under the GAEUS scoring system, videos are evaluated on a scale ranging from a minimum of one to a maximum of 75 points. Scores between 0–25 are classified as minimally useful, 25–50 as moderately useful, and 50–75 as highly useful. The videos included in the study were independently analyzed using GQS and M-DISCERN by two anesthesiologists with ten years of experience. The scores assigned by the two independent researchers were evaluated using Cohen's kappa test

to assess inter-rater reliability. Researchers independently evaluated the videos and assigned GQS, M-DISCERN, and GAEUS scores for each video. The final GQS, M-DISCERN, and GAEUS scores for each video were determined by averaging the assigned scores.

### Primary and secondary outcome measures

The primary outcome measure of our study was determined to be the GAEUS scores. The secondary outcome measures were identified as the GEACS, GQS, and M-DISCERN scores.

### Statistical analysis

Data analysis was performed using IBM SPSS Statistics 25 software (IBM Corp., Armonk, NY, USA). Continuous variables, such as video duration, number of likes, number of comments, time since publication, number of views, view rate, M-DISCERN, GQS, and GEACS scores, were presented as both mean ± standard deviation and median with interquartile range. The video source, uploader's occupation, and upload country, along with all categorical variables under the Preoperative Management, Intraoperative Management, and Postoperative Management categories, were presented as frequencies and percentages. For the reliability analysis of the study, Cronbach's alpha value was calculated and found to be 0.894, showing high consistency. For validity and reliability assessments, skewness and kurtosis analyses were performed; the accepted range for normality was determined as ±1.5. The maximum value obtained in our results was −1.076, indicating that the scales were objectively valid and reliable. To assess the consistency of video ratings given by the observers, Cohen's kappa coefficient was calculated. The results showed a kappa value of 0.758 for the GQS score and 0.889 for the M-DISCERN score, indicating strong inter-rater reliability for both measures. According to the Kendall's tau correlation performed, the GQS-usefulness score (r:0.367; p:0.000) and MDO-usefulness score (r:0.380; p:0.000) were statistically significant and highly correlated. Descriptive analyses were initiated. Mann–Whitney U test was used to compare group differences between continuous variables. Tukey's test was used for the analysis of within-group changes over time. Chi-square test was used for comparisons between categorical variables and groups. In all analyses, $p < 0.05$ was considered statistically significant.

## RESULTS

Initially, 100 videos related to geriatric anesthesia and 83 videos related to anesthesia in the elderly were found on YouTube. The study began with 183 videos; however, 96 videos were excluded from the study due to 20 shorts, 11 advertisements, nine non-English videos, seven unrelated titles, six veterinary-related videos, two voiceless videos, and 41 duplicates. The remaining 87 videos were examined in detail (Fig. 1).

   In our study, the descriptive characteristics of YouTube videos on anesthesia for geriatric and elderly patients were presented in Table 1. According to the results of our research, a significant portion of the videos, 48.3% (42 videos), was created by personal blogs. Content uploaded by educational institutions accounted for 36.8% (32 videos), while videos uploaded by healthcare facilities comprised 14.9% (13 videos). Most of the information in the videos was provided by doctors (94.3%, 82 videos).
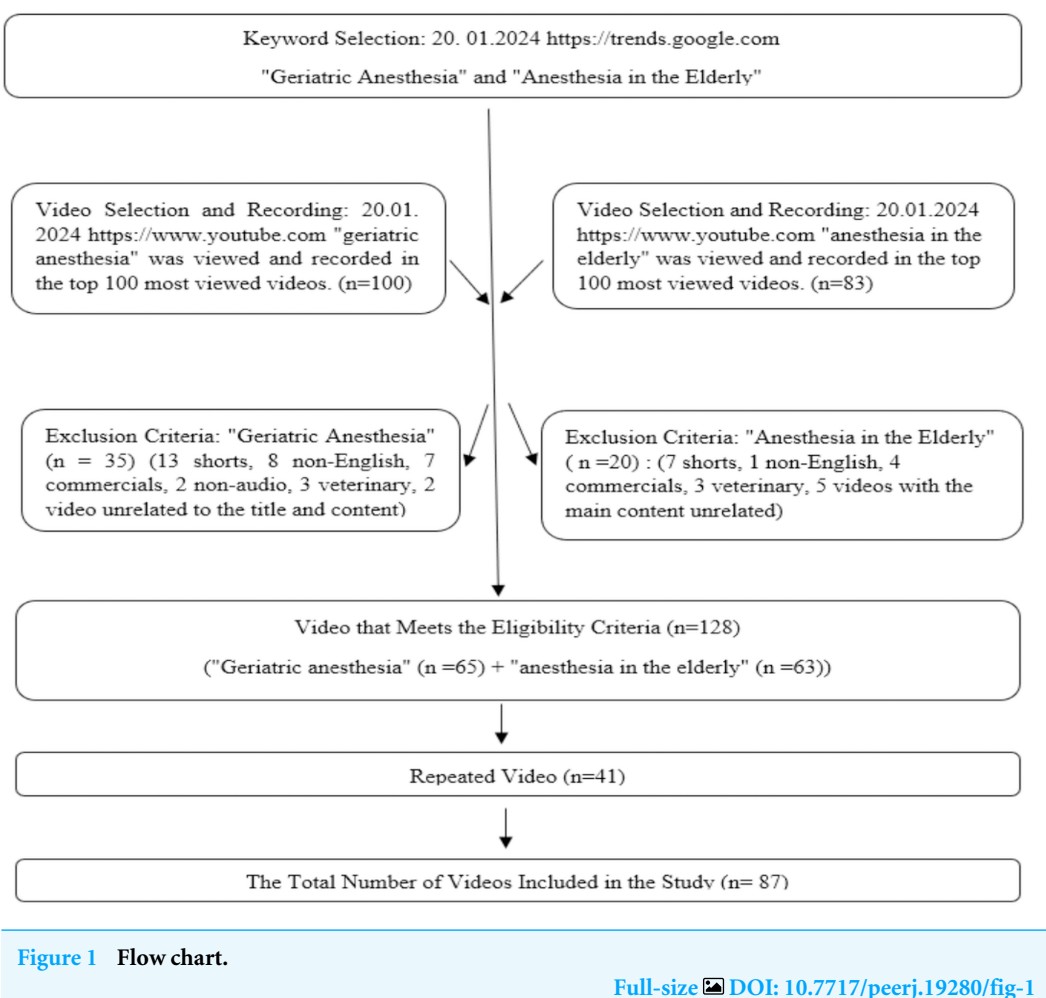

**Figure 1  Flow chart.**

Regarding the geographical distribution of the videos, the highest proportion came from unknown sources at 33.3% (29 videos), while the most videos were uploaded from the United States, accounting for 32.2% (28 videos).

The average duration of the examined videos was 35.42 min, with a median duration of 15 min (minimum 1 min, maximum 134 min). The average number of likes was 114.93, while the median number of likes was seven. The average duration that the videos remained on the platform was measured at 1139.03 days, with a median of 908 days. In terms of quality and reliability assessment, the GQS score averaged 3.34 with a median of three (range 1–5), and the M-DISCERN score averaged 3.37 with a median of 3.50 (range 1-5). The GEACS score averaged 11.95 with a median value of nine (range 0–58), while the GAEUS score averaged 15.30 with a median value of 12.5 (range 2–63) (Fig. 2).

In our study, thirty-five themes were evaluated under three main topic headings related to anesthesia videos for geriatric and elderly patients on YouTube (Table 2). In preoperative management, a total of nine sub-themes were addressed. According to the content distribution, healthcare institutions produced approximately 14.7% of the content with 22 entries, educational institutions contributed 48.0% with 72 entries, and personal

**Table 1 Descriptive characteristics of anesthesia video content in geriatric and elderly patients on YouTube.** Data points indicate averages calculated across all included videos, while additional metrics highlight variability within the dataset.

| Content information of the videos | Features | N (%) | |
|---|---|---|---|
| Video source | Health institution | 13 (14.9%) | |
| | Educational institution | 32 (36.8%) | |
| | Personal/Blog | 42 (48.3%) | |
| The profession of the uploader | Doctor | 82 (94.3%) | |
| | Other | 5 (5.7%) | |
| Uploaded country | USA | 28 (32.2%) | |
| | India | 12 (13.8%) | |
| | Other | 18 (20.7%) | |
| | Unknown | 29 (33.3%) | |
| Target audience | Patient | 16 (18.4%) | |
| | Healthcare worker/doctor | 71 (81.6%) | |
| Technical information of the videos | Mean±s.s. | Median (min.-max.) | Total |
| Video duration (minutes) | 35.42 ± 40.38 | 15 (1-134) | 3,082 min |
| Number of likes | 114.93 ± 519.34 | 7 (0-4,500) | 9,998 |
| Number of comments | 12.01 ± 83.51 | 0 (0-774) | 14,058 |
| The time elapsed from the day of publication to today (day) | 1,139.03 ± 909.55 | 908 (12-4,552) | 99,096 day |
| Number of views | 13,112.86 ± 59,477.37 | 325 (8-498.301) | 1,140,819 |
| View rate (number of views/time spent) | 6.59 ± 26.84 | 0.46 (0.02-206.96) | 520.87 |
| M-DISCERN | 3.37 ± 1.17 | 3.50 (1-5) | |
| GQS | 3.34 ± 1.16 | 3 (1-5) | |
| GEACS (1-70) | 11.95 ± 11.29 | 9 (0-58) | |
| GAEUS (2 –75) | 15.30 ± 11.67 | 12.5 (2-63) | |

**Notes.**

Abbreviations: USA, United States of America; M-DISCERN, Modified DISCERN Scale; GQS, Global Quality Scale; GEACS, Geriatric and Elderly Anesthesia Content Scale; GAEUS, Geriatric and Elderly Anesthesia Usefulness Score; N, Number of videos; SD, Standart Deviation.

blogs accounted for 38.7% with 58 entries. During this period, the topic theme with the most information across all sources was pathophysiology. Educational institutions provided significantly higher information on the ASA score and general health assessment topic compared to other sources ($p = 0.048$).

In intraoperative management, seventeen sub-themes were examined. In the content distribution, healthcare institutions contributed 13.8% with 34 entries, educational institutions contributed 41.5% with 102 entries, and personal blogs accounted for 52.9% with 130 entries. The topic theme with the most information across all sources during this period was anesthesia management. Educational institutions provided significantly higher information on the endotracheal intubation topic compared to other sources ($p = 0.024$).

In postoperative management, nine different sub-themes were analyzed. Healthcare institutions presented 12.2% with 14 entries, educational institutions presented 45.2%
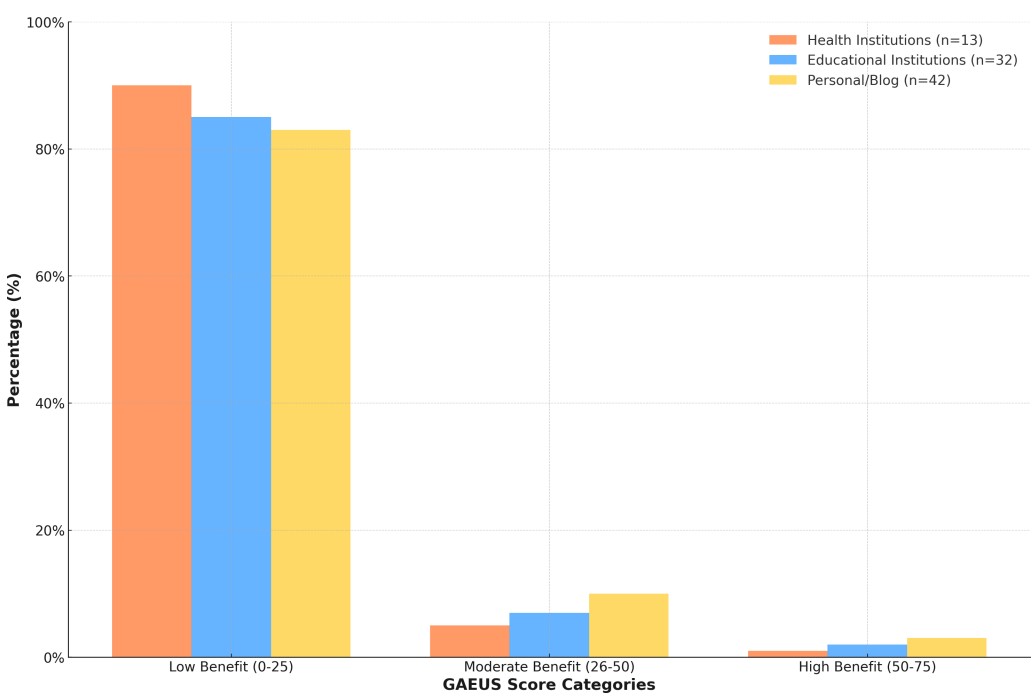

**Figure 2** Distribution of the usefulness of you tube videos related to geriatric anesthesia according to GEACS scores.

with 52 entries, and personal blogs presented 44.3% with 51 entries. During this period, the topic with the most information from educational institutions was postoperative delirium, with 37.5% (12 entries), and from personal blogs, it was also postoperative delirium with 35.7% (15 entries); from healthcare institutions, the topic with the most information was postoperative cognitive dysfunction at 30.8% (four entries).

In the analysis of the usefulness scores of videos related to anesthesia practices for geriatric and elderly patients, healthcare institutions provided content in the low usefulness category at 92.3%, educational institutions at 87.5%, and personal blogs at 83.3%. In the moderate usefulness category, healthcare institutions provided 7.7%, educational institutions 9.4%, and personal blogs 14.3%. In the high usefulness category, educational institutions and personal blogs provided very low content (3.1% and 2.4%, respectively), while healthcare institutions provided no content at all. No statistically significant difference was found between the GAEUS scores of the videos based on the sources ($p = 0.899$).

The usefulness scores of video content on anesthesia topics for geriatric and elderly patients on YouTube were compared in Table 3. The average duration of videos providing low usefulness was determined to be 31.59 min (range 5–44), while the average duration of videos providing moderate and high usefulness was 59.37 min (range 19.44–119.05). This difference in duration was statistically significant ($U = 2.569$, $P = 0.010$).

Videos providing low usefulness were published an average of 1,182.37 days ago, while those providing moderate and high usefulness were published an average of 868.17 days ago; however, this difference in duration was not statistically significant ($U = 0.659$,

**Table 2  Comparison of themes and usefulness scores of anesthesia videos in geriatric and elderly patients on YouTube according to the video source.** The table compares the main and sub-themes as well as usefulness scores of anesthesiarelated YouTube videos for geriatric and elderly patients, categorized by the source of the videos. Each data point reflects the proportion of videos addressing specific themes within the preoperative, intraoperative, and postoperative periods, along with their usefulness scores.

| Main theme | Sub-theme | Health institutions (*n* = 13) | Educational institutions (*n* = 32) | Personal/ Blog (*n* = 42) | *p* |
|---|---|---|---|---|---|
| Preoperative management | Definition epidemiology | 3 (23.1%) | 8 (25%) | 12 (28.2%) | 0.901 |
| | Pathophysiology | 5 (38.5%) | 18 (56.3%) | 18 (42.9%) | 0.413 |
| | Fragility | 4 (30.8%) | 11 (34.4%) | 6 (14.3%) | 0.112 |
| | ASA Score - General health values assessment | 0 | 5 (15.6%) | 1 (2.4%) | 0.048 |
| | Nutrition | 3 (23.1%) | 4 (12.5%) | 2 (4.8%) | 0.146 |
| | Concomitant diseases or comorbidity | 3 (23.1%) | 14 (43.8%) | 15 (35.7%) | 0.419 |
| | Substance use | 1 (7.7%) | 1 (3.1%) | 0 | 0.250 |
| | Polypharmacy | 2 (15.4%) | 6 (18.8%) | 4 (9.5%) | 0.514 |
| | Depression | 1 (7.7%) | 3 (9.4%) | 3 (7.1%) | 0.939 |
| Introperative management | Anesthesia management | 9 (69.2%) | 22 (68.8%) | 36 (85.7%) | 0.176 |
| | General anesthesia | 7 (53.8%) | 17 (53.1%) | 29 (69%) | 0.324 |
| | Spinal anesthesia | 3 (23.1%) | 4 (12.5%) | 6 (14.3%) | 0.657 |
| | Epidural anesthesia | 1 (7.7%) | 2 (6.3%) | 3 (7.1%) | 0.981 |
| | Peripheral nerve block | 2 (15.4%) | 5 (15.6%) | 4 (9.5%) | 0.699 |
| | Endotracheal Entubation | 0 | 7 (21.9%) | 2 (4.8%) | 0.024 |
| | Bag-valve-mask ventilation | 2 (15.4%) | 8 (25%) | 10 (23.8%) | 0.773 |
| | Extubation | 1 (7.7%) | 1 (7.7%) | 2 (4.8%) | 0.801 |
| | Monitoring | 5 (38.5%) | 19 (59.4%) | 26 (61.9%) | 0.315 |
| | Premedication | 0 | 4 (12.5%) | 6 (14.3%) | 0.360 |
| | Induction | 0 | 3 (9.4%) | 3 (7.1%) | 0.529 |
| | Anesthetic agents | 0 | 5 (15.6%) | 4 (9.5%) | 0.288 |
| | Neuromuscular blockers and their antagonists | 0 | 2 (6.3%) | 5 (11.9%) | 0.346 |
| | Other pharmacological agents | 4 (30.8%) | 11 (34.4%) | 22 (52.4%) | 0.195 |
| | Position | 0 | 4 (12.5%) | 9 (21.9%) | 0.148 |
| | Fluid management | 0 | 6 (18.8%) | 8 (19%) | 0.231 |
| | Blood transfusion | 0 | 2 (6.3%) | 1 (2.4%) | 0.506 |
| Postoperative management | Respiratory complications | 3 (23.1%) | 10 (31.3%) | 11 (26.2%) | 0.824 |
| | Nausea and vomiting | 0 | 2 (6.3%) | 1 (2.4%) | 0.506 |
| | Hypothermia | 1 (7.7%) | 4 (12.5%) | 5 (11.9%) | 0.894 |
| | Mobilization | 1 (7.7%) | 1 (3.1%) | 1 (2.4%) | 0.651 |
| | Physiotherapy | 2 (15.4%) | 5 (15.6%) | 1 (2.4%) | 0.105 |
| | Analgesia and postoperative pain | 1 (7.7%) | 7 (21.9%) | 10 (23.8%) | 0.446 |
| | Long-term follow-up and management of chronic conditions | 0 | 2 (6.3%) | 1 (2.4%) | 0.506 |
| | Postoperative delirium | 2 (15.4%) | 12 (37.5%) | 15 (35.7%) | 0.326 |
| | Postoperative cognitive dysfunction | 4 (30.8%) | 7 (21.9%) | 15 (35.7%) | 0.435 |

**Table 2** (*continued*)

| Main theme | Sub-theme | Health institutions (*n* = 13) | Educational institutions (*n* = 32) | Personal/ Blog (*n* = 42) | *p* |
|---|---|---|---|---|---|
| Total | 35 | 70 (13.08%) | 226 (42.24%) | 239 (44.67%) | |
| GAEUS (0 25) low benefit | | 12 (92.3%) | 28 (87.5%) | 35 (83.3%) | |
| GAEUS (26-50) moderate benefit | | 1 (7.7%) | 3 (9.4%) | 6 (14.3%) | 0.899 |
| GAEUS (50 75) high benefit | | 0 | 1 (3.1%) | 1 (2.4%) | |

**Notes.**
Chi-Square test.
Abbreviations: ASA, American Society of Anesthesiologists; GAEUS, Geriatric and Elderly Anesthesia Usefulness Score.

**Table 3  Comparison of anesthesia videos in geriatric and elderly patients on YouTube according to usefulness scores.** Each data point represents the median values of technical specifications for anesthesiarelated videos categorized by their usefulness scores.

| Technical specifications | GAEUS (0-25) low benefit | GAEUS (26-75) medium and high benefit | U | P |
|---|---|---|---|---|
| Video Duration (Minutes) | 13 (5-44) | 38 (19.44-119.05) | 2.569 | 0.010 |
| The time elapsed from the day of publication to today (day) | 1,182.37 (605-1,503) | 868.17 (647-1,056) | 0.659 | 0.510 |
| Number of views | 296 (99-2,016) | 399 (184.50-651.75) | 0.222 | 0.825 |
| View rate (number of views/time spent) | 0.44 (0.11-1.89) | 0.48 (0.28-0.76) | 0.191 | 0.848 |
| Number of likes | 6 (1-24) | 9 (5.25-13.25) | 0.735 | 0.463 |
| Number of comments | 0 (0-1) | 0 (0-1.75) | 0.573 | 0.567 |
| M DISCERN | 3 (2-4.5) | 4 (3.13-4.5) | 1.352 | 0.177 |

**Notes.**
Kruskal Wallis H Test test in analysis of countries.
Abbreviations: M-DISCERN, Modified DISCERN; GQS, Global Quality Scale; GEACS, Geriatric and Elderly Anesthesia Content Scale; GAEUS, Geriatric and Elderly Anesthesia Usefulness Score.

$P = 0.510$). Videos providing low usefulness were viewed an average of 15,052 times, while those providing moderate and high usefulness were viewed an average of 990.33 times; the difference in view counts was not statistically significant ($U = 0.222$, $P = 0.825$).

In terms of view rate, videos providing low usefulness were measured at an average of 7.58, while this rate for videos providing moderate and high usefulness was determined to be 1.07; however, this difference was not statistically significant ($U = 0.191$, $P = 0.848$). Videos providing low usefulness received an average of 129.99 likes, while those providing moderate and high usefulness received an average of 20.83 likes; this difference in the number of likes was also not statistically significant ($U = 0.735$, $P = 0.463$).

Regarding the number of comments, videos providing low usefulness had an average of 13.73 comments, while the average for videos providing moderate and high usefulness was 1.25; this difference was not statistically significant ($U = 0.573$, $P = 0.567$). The average M-DISCERN score for videos providing low usefulness was measured at 3.31, while

**Table 4 Comparison of quality, reliability, scope and usefulness of video content on anesthesia in geriatric and elderly patients on youtube according to the uploader's source.** Each data point reflects the median scores of quality, reliability, scope, and usefulness metrics for anesthesia-related YouTube videos targeting geriatric and elderly patients, grouped by the uploader's source.

| | Health institution ($n = 13$) | Educational institution ($n = 32$) | Personal/Blog ($n = 42$) | H | P |
|---|---|---|---|---|---|
| GQS | 3 (2-5) | 4 (3-4.88) | 3 (2-4) | 3.589 | 0.166 |
| M DISCERN | 3 (2-5) | 4 (3-4.5) | 3 (2-4) | 4.667 | 0.097 |
| GEACS | 5 (4-14) | 8.50 (2.5-13.75) | 10.5 (6-16.50) | 30.926 | 0.000 |
| GAEUS | 9.50 (7.5-17.5) | 12.5 (7-18) | 13.75 (8-20) | 30.833 | 0.000 |

**Notes.**

Mann Whitney U test in analysis of continents.

Abbreviations: M-DISCERN, Modified DISCERN; GQS, Global Quality Scale; GEACS, Geriatric and Elderly Anesthesia Content Scale; GAEUS, Geriatric and Elderly Anesthesia Usefulness Score.

for videos providing moderate and high usefulness, it was 3.83; however, no statistical difference was observed ($U = 1.352$, $P = 0.177$).

A comparison of video content on anesthesia topics for geriatric and elderly patients on YouTube was conducted based on the uploader source (Table 4). GQS for videos from healthcare institutions was three (ranging from two to five), while videos from educational institutions received a median score of four (ranging from three to 4.88), and videos from personal blogs scored three (ranging from two to four). The GQS scores of the videos did not show a statistically significant difference based on the uploader source ($p = 0.166$).

The M-DISCERN score revealed that healthcare institutions and personal/other sources had similar median scores (3), while educational institutions achieved a higher median value (4). However, the differences in M-DISCERN scores were not statistically significant ($p = 0.097$).

For videos from healthcare institutions, the median score for the GEACS was determined to be five (range 4–14), while for educational institutions it was 8.50 (range 2.5–13.75), and for personal/other sources, it was 10.5 (range 6–16.50). This difference was found to be statistically significant ($p = 0.000$).

For the GAEUS, the median usefulness score for videos from healthcare institutions was 9.50 (range 7.5–17.5), from educational institutions it was 12.5 (range 7–18), and from personal blogs it was 13.75 (range 8–20). These differences were statistically significant ($p = 0.000$).

# DISCUSSION

Our study is, to our knowledge, the first to evaluate the quality, reliability, content, and usefulness of anesthesia videos for geriatric and elderly patients on YouTube. In our study, we hypothesized that the anesthesia videos for geriatric and elderly patients on YouTube would be of moderate quality, reliability, content, and usefulness but would show significant differences depending on the sources. We assessed the usefulness of the videos using the GAEUS scoring system and found that they generally exhibited low usefulness. The highest GAEUS scores were obtained from personal blogs, followed by educational institutions and

healthcare organizations. The quality of the videos was measured using the GQS and their reliability was assessed with the M-DISCERN scale. Contrary to our hypothesis, we found no significant differences between the sources, as both scales indicated that the videos were of moderate quality. When measuring the thematic content scope with the GEACS, we determined that the broadest content was created by personal blogs. The most frequently addressed theme across all sources was anesthesia methods. The themes of ASA scores and general health assessments provided significantly more information in educational institutions compared to other sources.

Our study comprehensively analyzed 87 YouTube videos that have been viewed a total of 1.14 million times and contain about 51 h of content. When we look at the geographical distribution of video sources, the most content was created from the USA in our study and showed a similar distribution to the literature (*Parmar et al., 2023*). In our research, although similar results were obtained with the literature in terms of the educational usefulness of videos, the quality and reliability scores of our videos were found to be higher than the literature. Similar studies in the literature may explain the fact that although videos are mostly based on personal blog sources, our quality and reliability scores are high due to the fact that the total of videos produced by health institutions and educational groups in our study is higher than personal blogs and these institutions produce higher quality content (*Javidan et al., 2023*).

In our research, videos prepared mostly by doctors and health professionals achieved high quality scores above what is expected in the literature. We think that the fact that videos from other sources, such as personal blogs, also received high GAEUS scores may have been due to the higher quality of these contents than expected. In this context, we think that the fact that the quality and reliability scores of our videos are higher than the literature can be associated with the high degree of professionalism of the content presented. Although the educational usefulness of our videos is similar to that reported in the literature, the fact that their overall quality and reliability levels are lower may be due to our specially developed and detailed GEACS scoring system, which evaluates 35 different themes separately. This system measures the educational depth and content breadth of videos according to the number of themes. Consistent with studies in the literature, our videos were found to have low topic content and resulted in low educational usefulness scores (*Osman et al., 2022*). Previous research has shown that viewers' attention spans begin to decline after the fourth minute of a video, with significant reductions occurring after the sixth minute (*Guo, Kim & Rubin, 2014*). Other studies argue that videos should be kept under ten minutes to maximize viewer attention (*Lagerstrom, Johanes & Ponsukcharoen, 2015*). Considering viewers' attention spans, our study found a significant relationship between video duration and usefulness scores. An increase in usefulness scores with longer video durations has also been observed in previous studies (*Czerwonka et al., 2023*). Videos featuring expert speakers tend to receive more positive engagement and are shared more frequently. Additionally, previous studies have shown that higher-quality videos are generally longer in duration and that physician-led videos are associated with higher quality (*Kirkpatrick et al., 2021*; *Lock & Baker, 2022*). We think that by increasing the duration of anesthesia videos in geriatric and elderly patients, the scope of subject

content can be expanded, so their usefulness can increase. We did not find a relationship between viewer interactions, such as the number of views, likes and comments of videos, and the usefulness of videos. This result supports previous studies (*Macleod et al., 2015*). As a result, we also think that these audience interactions should not be used for usefulness.

In our study, the most frequently covered themes among all video sources were anesthesia management and general anesthesia, while regional anesthesia and especially intubation were less frequently covered. On the other hand, a study conducted on YouTube shows that the topics related to regional anesthesia, intubation and general anesthesia are examined the most in perioperative anesthesia videos. The frequency of themes in our study does not align with the titles more commonly used in other studies analyzing YouTube videos (*Nelms et al., 2024*). This discrepancy may be due to the low popularity of anesthesia videos aimed at geriatric and elderly patients. Although we selected the videos we took into the study from the most watched videos, the viewing rates used in many studies as an indicator of popularity in the literature are quite low in our study (*Kwak et al., 2022*). Additionally, a recent study showed that aging and death were the least popular topics among 10 major global health education themes. These results supported our theory that the popularity of anesthesia videos is low in geriatric and elderly patients (*Campbell & Rudan, 2020*).

With the aging population, the difficulties and complications faced by anesthesiologists are increasing. Elderly patients often face various health problems, such as multiple system disorders. This makes it mandatory to prepare customized anesthesia plans according to the individual needs of patients (*Van Zundert, Gatt & Van Zundert, 2023*). Many studies emphasize the importance of implementing the published guidelines for the perioperative care of the elderly (*Birkelbach et al., 2019*). However, various studies show that compliance with these guidelines is low (*Clark, Bennett & Foo, 2022*). In particular, video content providers should continue to produce more video content aimed at both healthcare professionals and patients based on current guidelines on disseminating accurate information about geriatric and elderly patients. We think that YouTube, which is a particularly popular video sharing site, can make an important contribution to meeting the need for useful information for this group. In addition, we also agree with the views that the academic incentives proposed in previous studies can contribute to the production of high-quality content on YouTube, and it may be useful to create a peer-reviewed section to disseminate accurate information (*Javidan et al., 2023*).

Our research has some important limitations. Firstly, the usefulness scoring system that we use to evaluate videos is not a generally accepted or verified method and is largely subjective. Second, the keywords we used in the video search process were assumed to be what a typical user would prefer, which could create potential bias. Because different users or situations may use various keywords and this may affect the results obtained. Third, our study only focused on English content, which may result in ignoring relevant videos in other languages. In addition, videos are sorted by the number of views on YouTube™, these ranking criteria may affect the results obtained. Additionally, our study did not account for the demographic characteristics of video viewers, which may influence how the content is perceived and utilized. Factors such as age, educational background, and level of medical knowledge can affect how viewers interpret and engage with the videos. The absence of

these demographic data limits our ability to determine the relevance and impact of the content for different audiences.

Furthermore, the scoring criteria used to evaluate video quality, reliability, and usefulness inherently involve a degree of subjectivity. While our assessment was conducted systematically, the lack of a universally validated and widely accepted scoring system may introduce potential bias. Future studies could benefit from incorporating standardized evaluation tools to enhance reproducibility and objectivity. Finally, YouTube search results are constantly changing; adding new videos or deconstructing existing ones can cause the results to change over time.

## CONCLUSION

In our study, we examined YouTube videos covering anesthesia topics for geriatric and elderly patients. The highest usefulness scores were obtained from personal blogs; however, all sources generally showed low usefulness. The overall quality of the videos was evaluated using theGQS, and their reliability was assessed with the M-DISCERN Scale. On both scales, the videos performed at a moderate level across all sources. These findings indicate the need for more comprehensive and informative resources on the YouTube platform, especially for the education of healthcare professionals and patients. To better meet the needs of elderly patients, the richness of content and educational value of the videos should be improved.

### Funding
The authors received no funding for this work.

### Competing Interests
The authors declare there are no competing interests.

### Author Contributions
- Turan Evran conceived and designed the experiments, performed the experiments, analyzed the data, prepared figures and/or tables, authored or reviewed drafts of the article, and approved the final draft.
- Seher İlhan performed the experiments, analyzed the data, authored or reviewed drafts of the article, and approved the final draft.

### Human Ethics
The following information was supplied relating to ethical approvals (i.e., approving body and any reference numbers):

Our research was approved by the Pamukkale University Faculty of Medicine Non-Interventional Clinical Research Ethics Committee on December 15, 2023 (No: E-60116787-020-462171).

## Data Availability

The raw data is available in the Supplemental File.

## Supplemental Information

Supplemental information for this article can be found online at http://dx.doi.org/10.7717/peerj.19280#supplemental-information.

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
