# Peer review of "Anesthesia videos in geriatric and elderly patients on YouTube: content, quality, reliability, and usefulness assessment"

_PeerJ, doi:10.7717/peerj.19280_

## Round 0.1 · original submission · Minor Revisions

Dear authors,
thank you for your submission. Some revisions are required before proceeding.
Please, refer to the reviewers comments for further details. Pay also attention to some language / syntax issues. And consider adding data information about the retrieval of some elements (such as number of likes or number of comments) as they are "time-sensitive". Revise tables / figures legends for completeness and highlight of key results.

Reviewer 1 ·

Basic reporting

The information provided in the background section is very nice.
line 94, "Today, health professionals, patients, and their families increasingly use the internet to find solutions to health problems, gain knowledge, and share their experiences." Here, add some sentences that show the reliability of the sources and what should be checked before direct application of the videos to patient management.
Do you think the duration of the video and experience of the video presenter matter in the delivery of required information? If yes, please add some evidence.

Experimental design

This study has substantial implications in anesthesia for elderly people.
The research questions are well-defined.
Methods are described with sufficient details.
line 193 Continuous variables were presented as mean ± standard deviation and median and interquartile range. Please specify for which datas you have used mean ± standard deviation, median, and interquartile range.

Validity of the findings

The findings of this research are meaningful and can serve as a foundation for future research in this area.
Conclusions are well stated, linked to the original research question, & limited to supporting results.

Additional comments

No comment.

Reviewer 2 ·

Basic reporting

1- Importance of the Topic: The importance of the topic has been sufficiently explained. Important points such as the increase in the aging population and the increase in the need for surgical intervention in elderly patients have been emphasized.
2- Literature Review: The literature review is up-to-date and covers relevant studies. Definitions from the World Health Organization and various research studies have been referenced.
3- Originality and Gap of the Study: The originality of the study and the gap in the literature have been clearly stated. It has been noted that there is no comprehensive evaluation of anesthesia videos for elderly and geriatric patients on YouTube in the literature.
4- Research Question or Hypothesis: The research question and hypothesis are clear and consistent. The aim is to evaluate the quality, reliability, content, and usefulness of anesthesia videos on YouTube from the perspective of healthcare professionals and patients.

Experimental design

The methodology section of your study appears to be well-structured and comprehensive. It includes detailed descriptions of the study design, keyword selection, video selection criteria, data collection techniques, and analysis methods.

Validity of the findings

The findings of this study provide a comprehensive analysis of the quality, reliability, and usefulness of anesthesia videos related to elderly and geriatric patients on YouTube. Here are some key points that support the adequacy of your findings:

Video Sources: It is noted that 48.3% of the videos were created by personal blogs. This indicates that your study covers a wide range of video sources.

Quality Assessment: The quality of the videos was evaluated using the GQS, with an average score of 3.34, indicating moderate quality. There was no significant difference according to the video source (p=0.166).

Reliability Assessment: The reliability was assessed using the M-DISCERN scale, with an average score of 3.37, indicating moderate reliability. There was no significant difference according to the video source (p=0.097).

Usefulness Assessment: The usefulness was evaluated using the GAEUS score, with an average score of 15.30, indicating low usefulness. However, there was a significant difference according to the video source (p=0.000).

Video Duration: The average duration of videos with low usefulness was 31.59 minutes, while the average duration of moderately and highly useful videos was 59.37 minutes. This duration difference was statistically significant (u=2.569, p=0.010).

These findings indicate that your study is methodologically sound and has sufficient data to evaluate the quality, reliability, and usefulness of anesthesia videos.

Additional comments

1.Inter-rater Reliability: Who are the evaluators of the videos? Are there independent evaluators? Has the agreement between evaluators been measured?
2. The country name in Table 1 should be corrected to USA.
3. In Table 2: In English, percentage numbers are written before the percent sign. For example, "23%" or "48.3%". Therefore, when writing in English, the percent sign should be placed before the number.
4. What is the meaning of ventilation; it should be written according to spelling rules.

·

Basic reporting

- The abstract is clear and effectively directs the reader to the focus of the research topic.
- The keywords should be presented in alphabetical order for consistency and ease of reference.
- The introduction broadly discusses the hypothesis regarding anesthesia-related YouTube videos. However, it would benefit from the inclusion of scientific literature or field-specific evidence related to elderly populations to strengthen its claims.
- Consider incorporating literature that explores the perceptions or feedback of elderly individuals regarding online health content. This would provide a more robust foundation for the study’s rationale and align it more closely with the stated objectives.

Experimental design

- A brief explanation of Google Trends and the criteria used for identifying the most viewed videos on "geriatric anesthesia" and "anesthesia in the elderly" should be included to enhance methodological clarity.
- The exclusion criteria for video selection should be explicitly stated to ensure transparency and reproducibility.
- A more detailed explanation of the scoring criteria employed in the methodology is necessary to facilitate replication and comprehension. (GQS, M-DISCERN, GEACS)

Validity of the findings

- The study’s findings are credible and supported by robust statistical analyses, with appropriate significance testing applied to assess differences in video quality, reliability,
and usefulness across different sources.
- The manuscript would be enhanced by the inclusion of visual tools, such as graphs or charts, to present the data more effectively.
- A deeper discussion comparing the findings with prior research is warranted. Additionally, the authors should explicitly address potential study limitations, such as the demographics of video viewers and the inherent subjectivity in scoring criteria.

Additional comments

- Typographical errors should be corrected throughout the manuscript to improve
readability.
- Consistency in word usage is essential; for example, "youtube" should be corrected to "YouTube" as it refers to a specific platform (line 95).

---

## Round 0.2 · accepted · Accept

Dear authors,
thank you for your cooperation. I am now accepting your manuscript for publication. Congratulations!

Reviewer 1 ·

Basic reporting

It is well-written.

Experimental design

The research design is nice.
The research questions are well defined.

Validity of the findings

The finding of this research is reproducible.

Reviewer 2 ·

Basic reporting

no comment

Experimental design

no comment

Validity of the findings

no comment

Additional comments

no comment

·

Basic reporting

Overall, the manuscript is well-written, with improved citation formatting that clearly presents the sources

Experimental design

The explanations regarding Google Trends, exclusion criteria, and scoring methods are clear and well-detailed

Validity of the findings

The study findings are strong, and the explanation of study limitations is sufficient. While demographic data is suggested, hopefully be explored in future research.ure research

Additional comments

Typographical errors have been well corrected